# Prostate-specific membrane antigen positron emission tomography compared to multiparametric MRI for prostate cancer diagnosis: a protocol for a systematic review and meta-analysis

Yi Zhao [1], Naomi Morka [2], Benjamin Scott S Simpson [3], Alex Freeman,[4] Alex Kirkham,[5] Daniel Kelly [6], Hayley C Whitaker,[7] Mark Emberton,[8,9] Joseph M Norris [3]

ME and JMN are joint senior authors.

For numbered affiliations see end of article.

**Correspondence to**
Yi Zhao;
yi.zhao18@imperial.ac.uk

## ABSTRACT

**Introduction** The introduction of multiparametric MRI (mpMRI) has improved almost every aspect of the prostate cancer diagnostic pathway. However, the novel imaging technique, prostate-specific membrane antigen positron emission tomography (PSMA PET) may have demonstrable accuracy in detecting and staging prostate cancer. Here, we describe a protocol for a systematic review and meta-analysis comparing mpMRI to PSMA PET for the diagnosis of suspected prostate cancer.

**Methods and analysis** A systematic search of MEDLINE, EMBASE, PubMed and Cochrane databases will be conducted. The Preferred Reporting Items for Systematic Reviews and Meta-Analyses guidelines will be followed for screening, data extraction, statistical analysis and reporting. Included papers will be full-text articles providing original data, written in English articles and comparing the use of PSMA PET with mpMRI in the diagnosis of prostate cancer. All studies published between July 1977 and March 2021 will be eligible for inclusion. Study bias and quality will be assessed using Quadas-2 score. To ensure the quality of the reporting of studies, this protocol is written following the Preferred Reporting Items for Systematic Review and Meta-Analysis Protocols 2015 checklist.

**Ethics and dissemination** Ethical approval will not be required for this systematic review. Findings will be disseminated through peer-reviewed publications and presentations at both national and international conferences.

**PROSPERO registration number** CRD42021239296.

### Strengths and limitations of this study

► This will be the first comprehensive systematic review and meta-analysis of the use of prostate-specific membrane antigen positron emission tomography (PSMA PET) and multiparametric MRI in the detection of prostate cancer, written in line with Preferred Reporting Items for Systematic Reviews and Meta-Analyses guidelines.

► This study will use comprehensive statistical methods to minimise the effect of heterogeneity in the literature on the determined results.

► The heterogeneity of study cohorts may limit the strength of conclusions drawn.

► There is likely to be a less extensive analysis of this focus due to the limited number of studies focusing on the primary diagnosis.

► Due to the recent introduction of PSMA PET as a mean of primary diagnosis, there may be limited long-term data on the clinical outcomes associated with this technology.

a result, the search for a more specific and sensitive imaging modality continues.

Prostate-specific membrane antigen positron emission tomography (PSMA PET) has primarily been used as a staging tool for established prostate cancer, however, there has been a recent surge in interest in using PSMA PET for diagnosis.[5] PSMA PET enables radionuclide imaging of PSMA, a type II transmembrane glycoprotein, overexpressed in the vast majority of prostate tumours,[6 7] and is known to correlate with high serum levels of prostate-specific antigen and a higher Gleason score, potentially making this a more targeted and specific marker of prostate cancer.[8] Conventionally, 68Ga-PSMA PET is coupled with either CT imaging or MRI,[8]

## BACKGROUND

Multiparametric MRI (mpMRI) is now routinely used in the prebiopsy risk stratification of suspected prostate cancer, affording significantly improved diagnostic accuracy over traditional techniques.[1 2]

However, approximately 10%–20% of clinically significant prostate tumours may be overlooked when using mpMRI,[3 4] and as



however, alternative radionuclides including, fluorine-18, have been proposed to improve imaging quality by overcoming the limitations exhibited when using the 68Ga radionuclides.[9]

Early PSMA PET diagnostic evaluation studies have shown promising results in the diagnosis of prostate cancer, particularly when compared with the current gold standard imaging approach, mpMRI. It is time to collate the growing evidence in this field, comparing the characteristics of these two imaging modalities, which has not been seen in the existing literature. Currently, existing literature focused on the diagnostic performance of PSMA PET and mpMRI in investigating the recurrence and metastasis of prostate cancer.[10 11] Other studies have focused on the individual diagnostic performance of either PSMA PET or mpMRI.[12]

This systematic review aims to analyse and summarise the diagnostic accuracies of PSMA PET and mpMRI for the first time, to potentially elucidate whether an alternative but reliable imaging modality can be determined for the detection of prostate cancer.

## METHODS AND ANALYSIS

This planned systematic review protocol is written in line with the Preferred Reporting Items for Systematic Review and Meta-Analysis Protocols 2015 checklist.[13] Included studies will undergo analysis and thematic synthesis to derive statistical data comparing the sensitivity, specificity, positive predictive value (PPV) and negative predictive value (NPV) between PSMA PET and mpMRI. The pooled sensitivities and specificities across the studies will be calculated first before deriving PPV and NPV values.

### Search methodology

A systematic search will be carried out using the MEDLINE, PubMed, EMBASE and Cochrane databases to retrieve all studies that contribute relevant evidence. The search will include Medical Subject Heading (MESH) terms and free text with appropriate Boolean operators. The following terms will be included in the search: "prostate", "'cancer", "diagnosis", as well as multiple synonyms for the term "PSMA PET" and "mpMRI" to account for differences in terminology. All eligible studies published between July 1977 and February 2021 will be uploaded to Rayyan, a semiautomated tool aimed at improving the speed and reporting accuracy of articles during the initial screening process.[14] To identify any missed studies or additional data, a further manual search of references in all included articles will be performed. In the case of absent or ambiguous data, the corresponding authors will be contacted directly for clarification.

### Study selection and data extraction

Three researchers will screen eligible studies independently and assess titles and abstracts for relevance. Should an article be considered eligible, the full text will be retrieved for further review of eligibility. Any

| Table 1 | Data collection items | |
|---------|-----------------------|---|
| Item no. | Data title | Data type |
| 1 | Year of publication | Study characteristic |
| 2 | Study authors | Study characteristic |
| 3 | Study design | Study characteristic |
| 4 | Patient population | Demographics |
| 5 | Number of participants | Demographics |
| 6 | mpMRI scoring scheme used | Methodology |
| 7 | Definition for clinically significant disease | Methodology |
| 8 | Definition for lesion visibility and invisibility | Methodology |
| 9 | Sample processing approach | Methodology |
| 10 | PSMA PET technique and radionuclide used | Methodology |
| 12 | Histopathological reference standard (including prostate biopsy and wholemount radical prostatectomy) | Methodology |
| 11 | Differential quantification of accuracies | Outcome |

mpMRI, multiparametric MRI.

disagreements between reviewers will be discussed until a consensus is reached, or a fourth reviewer will be consulted. All exclusions will be noted for further analysis and the reasons documented in detail to generate the Preferred Reporting Items for Systematic Reviews and Meta-Analyses flow diagram.

### Inclusion criteria

To be included in the analysis, studies must compare the diagnostic accuracies between PSMA PET and mpMRI modality for prostate cancer. Comparisons are focused on investigating sensitivities, specificities, PPVs and NPVs.

### Exclusion criteria

Correspondence articles, expert opinions, conference abstracts, review articles and case reports will be excluded. Selected studies must be written in the English language. Studies that do not make a direct comparison between PSMA PET and mpMRI, such as those investigating the combined accuracy in the diagnosis, will be excluded. Articles that solely focus on the diagnostic accuracy of PSMA PET or mpMRI alone will also be removed.

### Data extraction

All desired data will be collated in a dedicated datasheet. All reviewers will extract data independently.[15] Table 1 summarises the data to be collected.

### Endpoints

The primary endpoint will be statistically significant differences in quantitative measurements (eg, sensitivity, specificity, PPV and NPV) determining diagnostic accuracy between PSMA PET and mpMRI. We will focus on

key themes within the literature such as MRI scoring system (eg, Prostate Imaging-Reporting and Data System (PI-RADS), Likert and radiogenomic features) and the criteria used to define lesion visibility (PI-RADS or Likert score thresholds).

## Meta-analysis

The pooled quantitative diagnostic accuracy values between PSMA PET and mpMRI will be compared in the meta-analysis. In the first instance, sensitivity and specificity values will be retrieved or calculated. If studies do not provide these values they will be calculated from clinical tables or requested from study authors. In the instance of a substantial proportion of articles using any other metric, these values may also be retrieved and calculated separately. The distribution of untransformed, logit and double-arcsine transformed proportions will be assessed for normality using the Shapiro-Wilk test and density plots. The set of ratios that most resemble a normal distribution will be used for the combined analysis.

Inter-study variation and heterogeneity will be quantified through $I^2$ and as the true effect size for all studies are unlikely to be identical (due to methodological differences, institutional differences, interpretation differences) a random-effects model will be fitted for estimation and estimated with partial pooling. Once the model is fit, leave-one-out analyses (LOO) and accompanying diagnostic plots will be used to identify influential studies and will be quantified using externally studentised residuals, different in fits values, Cook's distances, covariance ratios, LOO estimates of the amount of heterogeneity, LOO values of the test statistics for heterogeneity, hat values and weights. Each study will be removed one at a time, and the summary proportions will be re-estimated based on the remaining n−1 studies. Studies with a statistically significant influence on the fitted model will be removed as outliers and the model re-fitted. Once heterogeneity has been minimised and outliers removed, summary estimates will be compared between PSMA PET and mpMRI. All data analysis and visualisation will be performed via the R statistical environment (V.3.6.1, 2019-07-05) using the 'mada' and 'mvmeta' packages.

## Risk of bias in individual studies

The Quadas-2 score will be used to assess bias and quality across the selected studies.[16] The scoring system is split into three main sections: selection, comparability and outcome. Within each section, there are sub-questions that assess the quality of research methodology at the study level. Three reviewers will be independently involved in this process with any disagreements settled by consensus. The outcome of the bias assessment will be used to influence data synthesis by assessing the applicability and reliability of the data produced. Studies may be excluded if found to be of low quality or suggesting high levels of bias. If they are included, appropriate commentaries will follow in the discussion. As this review is focused on the diagnostic accuracies of prostate cancer between PSMA PET and mpMRI, we will modify non-applicable sections of the scoring systems to reflect the nature of the evidence base and reduce reporting inaccuracy as reported previously.[15]

## Patient and public involvement

There will be no patient and public involvement in this study.

## DISCUSSION

Today, mpMRI is now widely adopted for prebiopsy risk stratification in men with suspected prostate cancer and has been incorporated into national and international prostate cancer guidelines.[17–19] However, the utility of alternative imaging modalities in this setting is underexplored. Through this systematic review, we aim to compare the diagnostic accuracies of a novel imaging technique, PSMA PET, against the new standard-of-care, mpMRI. The summation of evidence in this fledgeling field will aid in clinical and research decision-making when considering pooled accuracy in the diagnosis of primary prostate cancer, so benefitting patient care.

So far, the bulk of clinical evidence for PSMA PET has supported the use of this technique in the staging of prostate cancer. Indeed, PSMA PET/CT has been shown to independently predict treatment response to salvage radiation treatment, while PSMA PET/MRI has demonstrable accuracy in the staging of primary diagnosed prostate cancer.[20 21] Furthermore, the two most frequently used radionuclides, 68Ga or 18F, have been shown to have similar results in the staging of clinically significant prostate cancer.[22] However, only in recent years, has considerable attention been given to the potential benefits of PSMA PET in the diagnosis of suspected prostate cancer.

There may be several limitations in this review. There may be a small number of studies retrieved as the use of PSMA PET is relatively new. Moreover, there may be a small number of centres have access to PSMA PET technology, giving a limited generalisability of our study. The meta-analysis may be affected by the heterogeneity of results reported in each study. The different type of imaging modality used for both PSMA PET and mpMRI may act as confounders for the results.

This systematic review will combine the extant data, comparing the diagnostic performance of PSMA PET and mpMRI in this novel yet growing field, for the first time. The collation and analysis of these data will allow a better understanding of the differences in diagnostic utility between PSMA PET and mpMRI. Additionally, this process will also aid in clinical decision-making for prostate cancer diagnosis as well as potential areas for further research.

## Trial status

► Preliminary searches: started
► Piloting of the study selection process: started
► Formal screening: not started

- ▶ Data extraction: not started
- ▶ Risk of bias assessment: not started
- ▶ Data analysis: not started

## Draft search strategy

((((Prostat* AND (Cancer OR malignan* OR adeno-carcinoma OR lesion* OR Disease)) AND (PSMA OR "prostate-specific membrane antigen positron emission tomography")) AND (MR OR magnetic resonance imaging OR MP-MRI OR multi-parametric MRI OR multi-parametric magnetic resonance imaging OR multiparametric MRI OR "multiparametric magnetic resonance imaging")) AND Diagnosis).ti,ab

## ETHICS AND DISSEMINATION

Due to the nature of this review, there are no relevant ethical concerns and informed consent will not be required. The protocol and systematic review will be disseminated via a peer-reviewed journal.

**Author affiliations**
[1]Imperial College London, London, UK
[2]University College London Medical School, London, UK
[3]UCL Division of Surgery & Interventional Science, University College London, London, UK
[4]Department of Histopathology, University College Hospital London, London, UK
[5]Department of Radiology, University College London Hospitals NHS Foundation Trust, London, UK
[6]School of Healthcare Sciences, College of Biomedical and Life Sciences, Cardiff University, Cardiff, UK
[7]UCL Division of Surgery and Interventional Science, University College London, London, UK
[8]Division of Surgery and Interventional Sciences, University College London, London, UK
[9]Department of Urology, University College London Hospital, London, UK

**Contributors** The authors' contribution includes, but is not limited to, the following: NM, BSSS and JMN drafted the manuscript and created the study concept. AK, AF, DK, HCW and ME provided supervision and guidance during the study. All authors reviewed and approved the manuscript in its current form. YZ is the guarantor of this work.

**Funding** JMN is funded by the Medical Research Council (MRC) (MR/S00680X/1). BSSS is funded by the Royal Marsden Cancer Charity. ME receives research support from the UK's National Institute of Health Research (NIHR) UCLH/UCL Biochemical Research Centre.

**Competing interests** JMN receives funding from the MRC. Simpson receives funding from the Royal Marsden Cancer Charity. HCW receives funding from Prostate Cancer UK, the Urology Foundation and Rosetrees Trust. AK, AF and ME have stock interest in Nuada Medical. ME acts as a consultant, trainer and proctor to Sonatherm; Angiodynamics; Exact Imaging.

**Patient and public involvement** Patients and/or the public were not involved in the design, or conduct, or reporting, or dissemination plans of this research.

**Patient consent for publication** Not applicable.

**Provenance and peer review** Not commissioned; externally peer reviewed.

**ORCID iDs**
Yi Zhao http://orcid.org/0000-0002-4563-4344
Naomi Morka http://orcid.org/0000-0002-1925-7223
Benjamin Scott S Simpson http://orcid.org/0000-0003-3685-6110
Daniel Kelly http://orcid.org/0000-0002-1847-0655
Joseph M Norris http://orcid.org/0000-0003-2294-0303

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
