## [Reviewer comments · BMJ Open]

ARTICLE DETAILS

TITLE (PROVISIONAL)	Prostate-specific membrane antigen positron emission tomography compared to multiparametric magnetic resonance imaging for prostate cancer diagnosis: a protocol for a systematic review and meta-analysis
AUTHORS	Zhao, Yi; Morka, Naomi; Simpson, Benjamin Scott; Freeman, Alex; Kirkham, Alex; Kelly, Daniel; Whitaker, Hayley; Emberton, Mark; Norris, Joseph

VERSION 1 – REVIEW

REVIEWER	Leonard Marks University of California Los Angeles David Geffen School of Medicine, Urology
REVIEW RETURNED	12-May-2021

GENERAL COMMENTS	My only concern is the (apparently) limited number of publications dealing with this subject. Usually reviews like the proposed are based on a large body of peer-reviewed publications. I am only aware of a few, but perhaps the authors have knowledge of others that may be relevant. PSMA is of utmost interest, so I endorse the concept!
---

REVIEWER	Daan Nieboer Erasmus MC, Department of Public Health
REVIEW RETURNED	04-Oct-2021

GENERAL COMMENTS	The authors present a protocol for a systematic review and meta-analysis to compare the diagnostic accuracy of PSMA-PET and mpMRI for the diagnosis of prostate cancer. The protocol is well written and clear, however some methodological issues remain unaddressed. 1. Sensitivity and specificity should be analyzed using a bivariate meta-analysis model rather than two univariate meta-analyses of sensitivity and specificity. The r-packages meta and metafor are unable to fit these models.2. NPV and PPV are not only dependent on the test characteristics, but also on the prevalence of prostate cancer in the sample. Performing a meta-analysis of NPV and PPV is likely to introduce heterogeneity due to differences in prevalence of prostate cancer across studies. It may be better to perform a meta-analysis of sensitivity and specificity and subsequently calculate the PPV and NPV across a range of relevant prevalences.3. Would the Quadas-2 not be more appropriate for a systematic review comparing the diagnostic accuracy of two tests rather than the newcastle-ottowa scale?
--

	4. What is the reference test to which the PSMA-PET and mpMRI will be related? 5. Are there subgroup analyses/meta-regression planned to investigate possible sources of heterogeneity?
--	---

VERSION 1 – AUTHOR RESPONSE

Reviewer: 1

Dr. Leonard Marks, University of California Los Angeles David Geffen School of Medicine

Comments to the Author:

My only concern is the (apparently) limited number of publications dealing with this subject. Usually reviews like the proposed are based on a large body of peer-reviewed publications. I am only aware of a few, but perhaps the authors have knowledge of others that may be relevant. PSMA is of utmost interest, so I endorse the concept!

Thank you Dr. Marks. Since the FDA approval of using Gallium 68 PSMA-11 (<https://www.fda.gov/news-events/press-announcements/fda-approves-first-psma-targeted-pet-imaging-drug-men-prostate-cancer>) many literature has been published comparing the diagnostic performance of PSMA PET and mpMRI. At the time of literature research (Feb 2020) we have eluded more than 10 papers matching the inclusion criteria for meta-analysis and we will continue to invite papers for our analysis.

Reviewer: 2

Dr. Daan Nieboer, Erasmus MC

Comments to the Author:

The authors present a protocol for a systematic review and meta-analysis to compare the diagnostic accuracy of PSMA-PET and mpMRI for the diagnosis of prostate cancer. The protocol is well written and clear, however some methodological issues remain unaddressed.

1. Sensitivity and specificity should be analyzed using a bivariate meta-analysis model rather than two univariate meta-analyses of sensitivity and specificity. The r-packages meta and metafor are unable to fit these models.

We will be using “mada” for bivariate meta-analysis as stated in “mvmeta” package for R (<https://cran.r-project.org/web/packages/mvmeta/mvmeta.pdf>). This is reflected the Meta Analysis section

2. NPV and PPV are not only dependent on the test characteristics, but also on the prevalence of prostate cancer in the sample. Performing a meta-analysis of NPV and PPV is likely to introduce heterogeneity due to differences in prevalence of prostate cancer across studies. It may be better to perform a meta-analysis of sensitivity and specificity and subsequently calculate the PPV and NPV across a range of relevant prevalences.

We will aim to calculate the pooled sensitivity and specificity across the studies using the relevant statistical packages and then calculate the PPV and NPV values. The correction is highlighted on the 1st paragraph of METHODS and ANALYSIS

3. Would the Quadas-2 not be more appropriate for a systematic review comparing the diagnostic accuracy of two tests rather than the newcastle-ottowa scale?

We will be conducting Quadas-2 analysis for risk of bias. The correction is highlighted at the ‘Risk of Bias in individual studies’ section

4. What is the reference test to which the PSMA-PET and mpMRI will be related?

All papers have used histopathology as a gold standard diagnosis of prostate cancer and were used in evaluating Type I and Type II errors for both PSMA-PET and mpMRI. This is reflected in Table 1.

5. Are there subgroup analyses/meta-regression planned to investigate possible sources of heterogeneity?

There will be subgroup analysis employed to investigate possible sources of heterogeneity provided that there are enough studies in each subgroup. If not, we will mention this in the discussion for it to be done when enough studies accumulate.